# Ceramide Synthase 6 Maximizes p53 Function to Prevent Progeny Formation from Polyploid Giant Cancer Cells

**DOI:** 10.3390/cancers13092212

**Published:** 2021-05-05

**Authors:** Ping Lu, Shai White-Gilbertson, Gyda Beeson, Craig Beeson, Besim Ogretmen, James Norris, Christina Voelkel-Johnson

**Affiliations:** 1Department of Microbiology & Immunology, Medical University of South Carolina, Charleston, SC 29425, USA; lup@musc.edu (P.L.); whitesh@musc.edu (S.W.-G.); norrisjs@musc.edu (J.N.); 2Department of Drug Discovery and Biomedical Sciences, Medical University of South Carolina, Charleston, SC 29425, USA; beeson@musc.edu (G.B.); Beesonc@icloud.com (C.B.); 3Department of Biochemistry and Molecular Biology, Medical University of South Carolina, Charleston, SC 29425, USA; ogretmen@musc.edu

**Keywords:** sphingolipids: p53, polyploidy, ceramide, ceramide synthase, cancer

## Abstract

**Simple Summary:**

One mechanism that contributes to cancer recurrence is the ability of some malignant cells to temporarily halt cell division and accumulate multiple nuclei that are later released as progeny, which resume cell division. The release of progeny occurs via primitive cleavage and is highly dependent on the sphingolipid enzyme acid ceramidase but the role of sphingolipid metabolism in this process remains to be elucidated. This study highlights differences in sphingolipid metabolism between non-polyploid and polyploid cancer cells and shows that ceramide synthase 6, which preferentially generates C_16_-ceramide maximizes the ability of the tumor suppressor p53 to inhibit progeny formation in polyploid cancer cells. These results offer an explanation as to why non-cancerous polyploid cells, which express wildtype p53, do not generate progeny and suggest that cancer cells with deregulated p53 function pose a higher risk of evading therapy especially if enzymes that generate C_16_-ceramide are also dysregulated.

**Abstract:**

Polyploid giant cancer cells (PGCC) constitute a transiently senescent subpopulation of cancer cells that arises in response to stress. PGCC are capable of generating progeny via a primitive, cleavage-like cell division that is dependent on the sphingolipid enzyme acid ceramidase (ASAH1). The goal of this study was to understand differences in sphingolipid metabolism between non-polyploid and polyploid cancer cells to gain an understanding of the ASAH1-dependence in the PGCC population. Steady-state and flux analysis of sphingolipids did not support our initial hypothesis that the ASAH1 product sphingosine is rapidly converted into the pro-survival lipid sphingosine-1-phosphate. Instead, our results suggest that ASAH1 activity is important for preventing the accumulation of long chain ceramides such as C_16_-ceramide. We therefore determined how modulation of C_16_-ceramide, either through CerS6 or p53, a known PGCC suppressor and enhancer of CerS6-derived C_16_-ceramide, affected PGCC progeny formation. Co-expression of the CerS6 and p53 abrogated the ability of PGCC to form offspring, suggesting that the two genes form a positive feedback loop. CerS6 enhanced the effect of p53 by significantly increasing protein half-life. Our results support the idea that sphingolipid metabolism is of functional importance in PGCC and that targeting this signaling pathway has potential for clinical intervention.

## 1. Introduction

Cancer is routinely treated with anti-mitotic therapies, such as chemotherapy or radiation, which are intended to induce cell death in malignantly proliferative cells. Unfortunately, cancer cells that survive treatment have the potential to drive disease relapse and, due to their tendency to be more aggressive, metastatic, and resistant to therapy, this presents a clinical challenge. The process by which cancer cells survive the initial treatment and subsequently cause relapse, has recently received increased attention and a paradigm to elucidate the arc of events is emerging [1,2]. It has been shown that blocking mitosis drives a subpopulation of cancer cells to de-differentiate into an early, embryonic-like state resembling blastomeres [3]. While this likely offers numerous advantages, an immediate benefit is the re-activation of reproductive strategies other than mitosis. Pre-implantation embryonic cells, such as 2–8 cell blastomeres, are polyploid and divide through “primitive cleavage” rather than true mitosis [4]. Reversion to this stage allows cancer cells under treatment stress the physiological flexibility to circumvent mitotic catastrophe or apoptosis [5]. Instead of undergoing cell death these cancer cells adapt to stress by temporarily assuming a polyploid state that poises them to release mononuclear progeny by budding or bursting once the stress has resolved [5,6,7,8,9,10]. The polyploid giant cancer cells (PGCC) have now been recognized as a driver of recurrence and present a new focus of interest among cancer researchers and clinicians [7,11,12].

We recently identified the sphingolipid enzyme acid ceramidase (ASAH1) as the first molecular target for interfering with the generation of progeny from PGCC [13]. ASAH1 resides in the lysosome, where it hydrolyzes ceramides to generate sphingosine, which serves as a substrate for sphingosine kinase 1 or 2 (SphK1 and SphK2) that through the addition of a phosphate group generate sphingosine-1-phosphate (S1P) [13,14]. Ceramides and S1P are functionally opposed, with the former generally considered pro-apoptotic and the latter playing roles in survival, mitosis and angiogenesis. Therefore, the conversion of ceramide to sphingosine by ASAH1 is typically assumed to tilt the balance in the cell away from apoptosis and toward pro-survival signals. However, sphingosine can also be recycled back to ceramide by any one of six ceramide synthases, which through preferential substrate specificity generate ceramides with different fatty acyl side chains.

In this study, we sought to understand the fate of the ASAH1 product sphingosine in order to elucidate the roles of sphingolipids in PGCC progeny formation. Dogma suggests that adaptation to stress would most likely drive sphingolipid metabolism toward S1P, since this bioactive lipid has been associated with survival. However, our data did not support a role for increased S1P as a driver for PGCC progeny formation since inhibition of sphingosine kinases failed to prevent formation of PGCC progeny. Instead, our results suggest that reducing levels of C_16_-ceramide is important for PGCC progeny formation, which could be most efficiently prevented through co-expression of p53 and CerS6.

## 2. Materials and Methods

### 2.1. Cell Lines and Culture

PPC1 cells were a kind gift from Dr. Dean Tang (Roswell Park Comprehensive Cancer Center, Buffalo, NY, USA). MEL624 cells were a kindly provided by Dr. Michael Nishimura (Loyola University, Chicago, IL, USA). Cells were authenticated and regularly monitored to ensure cultures are mycoplasma-free. PPC1 cells expressing shRNA against CerS6 or GFP as control were previously generated by stable transfection with plasmids from Open Biosystems (Huntsville, AL, USA) [15]. All cells were maintained in RPMI1640 media (Cellgro, Manassas, VA, USA) supplemented with 10% heat-inactivated FBS (Cat # SH3007103 Hyclone, Omaha, NE, USA) and 1% antibiotic/antimycotic (Gibco, Marietta, GA, USA). Transfected cells were maintained in the presence of 5 μg/mL puromycin (Alfa Aesar, Ward Hill, MA, USA). Trypsin for subculturing was from Cellgro (Manassas, VA, USA). PGCC were generated through radiation stress (8Gy) or treatment with 5 nM docetaxel (LC Laboratories, Woburn, MA, USA). PGCC enrichment was achieved by filtration (20 micron filter, Pluriselect, El Cajon, CA, USA).

### 2.2. Reagents

D-erythro-sphingosine (^17^C-base, cat# 860640) and GT-11 (cat# 857395) were obtained from Avanti Polar Lipids (Alabaster, AL, USA). PF543 (Cat#17034, Cayman Chemical), ABC294640 was a kind gift from Dr. Charles Smith (Apogee Biosciences Corp, Hershey PA). LCL521 was obtained from the MUSC Lipids Core [16]. Cycloheximide was from Cayman Chemicals (cat# 601105, Ann Arbor, MI, USA).

### 2.3. Adenoviral Vectors and Transductions

All adenoviral vectors were replication-deficient and derived from human adenovirus type 5 (dE1/E3). Ad-CerS6 was custom generated by VectorBiolabs (Malvern, PA, USA). Pre-made Ad-Luc was also purchased from VectorBiolabs (cat#1000). Ad-p53 was a gift from Dr. Sunil Chada [17,18]. For transductions, cells were plated at 1 million/100mm dish, allowed to adhere overnight, and viral vectors were added at 1MOI (PPC1) or at 3MOI (MEL624) per construct. Ad-Luc served as control.

### 2.4. Immunoblotting

Cells were transduced with viral vectors and protein expression analyzed two days post-transduction. For the p53 half-life study, PPC1 cells stably expressing shRNA against GFP or CerS6 were plated a 7 × 10^4^ per well in a six-well plate, allowed to adhere overnight, and transduced with 1 MOI of Ad-p53 the next day. Two days post-transduction, cells were treated with 25 µg/mL cycloheximide for the indicated times and then scraped into PBS, pelleted by centrifugation and pellets lysed in RIPA buffer. Western blotting was performed as previously described [19]. Primary antibodies (1:1000) were p53 (cat# 2527, Cell Signaling, Danvers, MA, USA), CerS6 (cat# ab565820, Abcam, Cambridge, UK), and actin (cat# A2066, Sigma, St. Louis, MO) or GAPDH (Cat# sc32233, Santa Cruz, Dallas, TX). Secondary antibodies (1:5000) were obtained from Santa Cruz Biotechnology (Dallas, TX) (anti-mouse cat# sc-2005 and anti-rabbit cat# sc-2004).

### 2.5. LC/MS Analysis of Sphingolipids

Cells were plated at 8 × 10^5^ per 100 mm plate overnight and were either left untreated or irradiated with 8 Gy. LCL521 treatments (5 µM for 5 h) were done 3 days after irradiation when PGCC had formed (steady-state analysis). For flux studies, cells were switched to serum-free RPMI1640 on day 2, and cells treated with 10 µM GT-11 for one hour prior to addition of 2 µM d-erythro-sphingosine (^17^C-base) for 30 min. For studies with sphingosine kinase inhibitors, 1 million cells were plated in 100 mm plates overnight and incubated with inhibitors as indicated for 5 h. For analyses involving viral vectors, docetaxel was added at 5nM one day after viral transduction for 2 days. At endpoints, cells were scraped into PBS, pelleted, and pellets stored at −80C until lipidomics analysis by LC/MS [20].

Results obtained from the LC/MS analysis of sphingolipids are typically normalized to either phosphates, cell number, or protein. Due to the different physiology and cell size between parental cancer cells and their PGCC derivatives, normalization was complicated. First, PGCC have approximately eight times the DNA per cell (Appendix A [13], and are more aerobic (Appendix A–D [21]), which results in higher phosphate levels in PGCC samples (Appendix A [22]). Second, normalizing to cell number skews results due differences in cell size between parental cancer cells and their giant polyploid counterparts that have an increase in cellular material (Appendix A). Finally, while normalization to total protein per sample initially appeared promising (Appendix A), treatment with LCL521, even for durations as short as 5 h, resulted in loss of protein well before any loss of viability in PGCC only (Appendix A). The underlying reason for the decrease in protein remains to be determined but it precluded normalization using this method. Thus, in order to compare sphingolipids between parental cancer cells and PGCC, we analyzed the distribution of sphingoid bases in each sample, which stabilized readouts across experiments as shown in Appendix A–G. Therefore, lipidomics results between cancer cells and their PGCC counterpart are expressed as percent distribution within each experimental condition throughout this manuscript.

### 2.6. Cell Cycle Analysis

Flow cytometry to quantify polyploid cells was performed as previously described [13].

### 2.7. Analysis of Progeny Formation

Cells were seeded at 8 × 10^5^ on 100-mm plates overnight and irradiated the next day. After three days, PGCC were captured, plated sparsely on six-well plates and allowed to recover in the absence or presence of inhibitors. Inhibitors were refreshed every 3 days. For assays involving viral transductions, cells were plated 8 × 10^5^ on 100-mm plates overnight, transduced with viral vectors, and then treated with 5nM docetaxel. Two days after docetaxel treatment, PGCC were captured and plated as described above. At endpoints, colonies were fixed with ice-cold methanol for 20 min, air dried, stained with crystal violet, and visualized with a Zeiss Axiovert 200 microscope. Colonies containing at least 15 cells surrounding a PGCC were counted.

### 2.8. Statistics

ANOVA was used to evaluate overall differences across groups with multiple conditions. If no difference was detected, the entire group was denoted as not significant (‘ns’). If the ANOVA identified differences, Student’s *t*-test was used to further analyze specific comparisons of interest as shown in each figure. Two-sided testing was performed with α set to 0.05 for each *t*-test. Analysis was performed with Excel or GraphPad Prism 9 (San Diego, CA, USA). Each experiment shown was performed as at least three independent assays, typically in triplicate each time.

## 3. Results

### 3.1. The PGCC Ceramide-Sphingosine-S1P Circuit Is Similar but Not Identical to Parental Cancer Cells

We have previously shown that ASAH1 expression increases in PGCC and its activity is critical for PGCC progeny formation across cell lines from different types of cancer [13,14]. To better understand the specific dependence of PGCC on ASAH1, LC/MS analysis was performed to analyze sphingolipid profiles in parental PPC1 cells and their PGCC derivatives in the presence and absence of the ASAH1 inhibitor LCL521. As expected, treatment with LCL521 decreased the ASAH1 product sphingosine and its phosphorylated downstream metabolite S1P in both parental PPC1 cancer cells as well as the stress-induced PGCC (Figure 1A,B). In the absence of LCL521, S1P levels were comparable between parental and PGCC populations but PGCC had significantly lower levels of sphingosine (2.82 ± 0.93% vs. 9.21 ± 2.28% of total sphingoid bases) (Appendix A). We hypothesized that the reduced level of sphingosine detected in PGCC compared to parental cancer cells was due to increased activity of sphingosine kinases that use sphingosine to generate the pro-survival lipid S1P. We further hypothesized that the lack of difference in steady-state intracellular levels of S1P between parental cells and PGCC was due to secretion of S1P into the media or enhanced activity of S1P lyase, which degrades S1P.

To test this hypothesis, we performed flux experiments to trace how sphingosine with an artificial ^17^C-backbone (^17^C-Sph) is metabolized by PPC1 parental cells and PGCC. These experiments were performed in the presence and absence of GT-11 to inhibit the S1P lyase. Our results indicated that parental cells and PGCC metabolized the ^17^C-Sph at similar rates with equivalent amounts of the ^17^C-Sph detected unmodified in both the media and intracellular compartments after 30 min (Figure 1C). Parental cells and PGCC released similar levels of S1P into the media and while S1P accumulated to higher intracellular levels in the presence of GT-11, there was no preferential increase of S1P in PGCC (Figure 1D). Overall, sphingosine was primarily metabolized to C_16_-ceramide in both parental cells and PGCC, suggesting high activity of CerS5/6 in the salvage pathway but no statistically significant differences were observed across conditions for any of the ceramide species (Figure 1E).

Since the results did not support our hypothesis, we next examined differences in ceramide species under conditions when ASAH1 was active or inhibited. A key difference between parental cells and PGCC was that the latter had more total ceramide at baseline (97.0 ± 0.93% vs. 90.6 ± 2.28% of total sphingoid bases). Unlike in the flux analysis, where C_16_-ceramide emerged as the predominant species, the most abundant species at steady state was C_24_-ceramide in both parental cells and PGCC (Appendix A).

However, when ASAH1 was inhibited with LCL521, C_14_- C_16_- and C_18_-ceramides increased preferentially PGCC, which suggested a potential role for long chain ceramides in PGCC. (Figure 1F). The accumulation of C_16_-ceramide upon ASAH1 inhibition was of particular interest, given that the flux data showed that sphingosine is primarily metabolized into this species.

Since ceramide profiles at steady state did not reflect the flux analysis, we also analyzed sphingomyelins and found that C_16_-sphingomyelin was the most abundant species in PPC1 cells with the sphingomyelin profiles more closely resembled the flux analysis (Appendix A). Similarly, MEL624 melanoma cells had high levels of C_24_-ceramides at steady state, but the predominant sphingomyelin was the C_16_-species (Appendix A). Taken together, these results suggest that the activity of CerS5/6, which preferentially generate C_16_-ceramide, is high but that C_16_-ceramide does not accumulate intracellularly due to further metabolism in complex sphingolipids such as sphingomyelin.

### 3.2. PGCC Are Not Dependent on S1P for Survival or Reproduction

To functionally validate the data obtained by LC/MS analysis, we quantified the ability of PGCC to form progeny to address two important questions: (1) Does S1P play a specific role in PGCC despite demonstrating similarities to parental cells? (2) Is the ability of PGCC to generate progeny vulnerable to C_16_-ceramide accumulation? First, we determined the relative contribution of SphK1 and SphK2 to total S1P in PPC1 cells by evaluating how the SphK1 inhibitor PF543 and the SphK2 inhibitor ABC294640 (Opaganib, Yeliva^®^) impacted intracellular S1P levels. Figure 2A shows that the SphK1 inhibitor PF543 was highly effective at reducing intracellular S1P, while the SphK2 inhibitor ABC294640 had no significant effect, which suggested that SphK1 is preferentially responsible for the pool of S1P in PPC1 cells. Next, PGCC were treated with 10 µM PF543, 10 µM ABC294640 and their ability to generate progeny quantified through colony counting. As positive controls, we included 5 µM LCL-521 or 5 µM tamoxifen, which inhibit ASAH1 upstream of sphingosine kinases. As shown in Figure 2B, only inhibition of ASAH1, but not sphingosine kinases, prevented progeny formation. To ensure this effect was not unique to PPC1 prostate cancer cells, we confirmed these results in MEL624 melanoma cells (Figure 2C,D). In summary, our data suggest that PGCC are not dependent on SphK1 activity for progeny formation.

### 3.3. C_16_-Ceramide Can Be Modulated Directly via CerS6 or Indirectly via p53

Since we were unable to establish a role for S1P and sphingosine kinases in PGCC, we focused on the potential role of C_16_-ceramide in preventing progeny formation. C_16_-ceramide levels were modulated directly through knockdown or overexpression of CerS6 [23,24]. We also modulated C_16_-ceramide levels indirectly though p53. Loss of p53 is important for polyploidy and tumorigenicity but the role of sphingolipids in these processes is unknown [25,26,27]. The Krupenko laboratory recently identified a positive feedback loop in which C_16_-ceramide serves as an activator of p53 by stabilizing the protein through direct interaction with the DNA binding domain, and p53 in turn transcriptionally activates CerS6, which preferentially generates C_16_-ceramide [28]. PPC1, which are derived from PC3 cells, lack p53 protein expression due to loss of one allele and a truncating mutation in the other [29]. We hypothesized that expressing low, non-apoptotic levels of wildtype p53 could be used to indirectly increase intracellular C_16_-ceramide. To isolate the role of the p53/CerS6 feedback loop from any general effects of p53, we performed assays in parallel with targeting CerS6 by shRNA. Modulation of gene expression is shown in Figure 3A, and the corresponding C_16_-ceramide levels are quantified in Figure 3B, with the effects on all ceramides shown in Appendix A. Cells lacking CerS6 and p53 have the lowest levels of C_16_-ceramide, whereas cells expressing both genes have the highest levels of this ceramide species. Next, we confirmed that the virally transduced wildtype p53 is stabilized by C_16_-ceramide and found that the p53 half-life is dramatically shortened from over 60 min to under 15 min in the reduced C_16_-ceramide environment found in cells expressing the CerS6 shRNA (Figure 3C,D).

### 3.4. PGCC Progeny Formation Is Suppressed by p53 but Requires CerS6 for Maximal Efficacy

To understand the role of CerS6 in the absence or presence of p53, we exposed cells with differential expression in both genes to docetaxel stress and quantified the formation of polyploid cells via flow cytometry. The presence or absence of CerS6 alone did not impact the percentage of PGCC either at baseline or following docetaxel stress. In contrast, cells expressing p53 had a reduced percentage of PGCC at baseline and following stress relative to controls (Figure 4A,B). However, the fold-increase in polyploidy in response to docetaxel stress was not significantly different between cells lacking (2.8-fold) or expressing p53 (2.6-fold) (Figure 4A,B). To test the ability of PGCC to generate progeny, equal numbers of PGCC were plated and cultures observed for the formation of colonies. PGCC lacking p53 were able to robustly form colonies regardless of CerS6 status (Figure 4C). However, in the presence of p53, colony formation was influenced by CerS6 expression. PGCC expressing p53 but lacking CerS6 were able to generate progeny that established colonies, suggesting that p53 requires CerS6 for maximal suppression (Figure 4C,D).

We confirmed these results in MEL624 melanoma cells, which lack expression of both p53 and CerS6 (Figure 5A). C_16_-ceramide levels upon reconstitution of CerS6, p53, or both is shown in Figure 5B. Similar to PPC1 cells, CerS6 expressing cells generate more C_16_-ceramide but expression of CerS6 alone did not affect colony formation in melanoma cells (Figure 5C). Expression of p53 alone reduced the ability of PGCC to establish colonies but CerS6 was required for maximum suppression (Figure 5D).

## 4. Discussion

We previously showed that ASAH1 function is critical for PGCC progeny formation, and since ASAH1 inhibition preferentially impacts functions of PGCC but not the parental cancer cells, we hypothesized that these two populations of cancer cells have fundamental differences in their sphingolipid metabolism. In this study, we aimed to understand the fate of ASAH1-generated sphingosine to further elucidate the roles of sphingolipids in PGCC progeny formation. We initially analyzed sphingolipid profiles to identify differences between parental cancer cells and their polyploid derivatives in the absence and presence of the ASAH1 inhibitor LCL521. A key difference was that PGCC, despite higher ASAH1 expression, had relatively lower levels of the enzymatic product sphingosine (Figure 1A), which suggested the possibility that sphingosine is rapidly metabolized to S1P by sphingosine kinases [13]. Numerous studies have shown that S1P is associated with survival and drug resistance, which are two characteristics of PGCC [1,30]. However, our results did not support a role for S1P in PGCC. We determined that the majority of S1P in PPC1 cells was generated by SphK1 (Figure 2A) but flux assays failed to support increased S1P generation or secretion in PGCC compared to parental cells (Figure 1). Furthermore, inhibition of SphK1 did not affect colony formation in PGCC (Figure 2B,D). Treatment with the SphK2 inhibitor also failed to affect PGCC colony formation but additional studies using genetic ablation of SphK2 would be required to rule out a role for this kinase, since a relative low amount of S1P appears to be derived from SphK2. The SphK2 inhibitor may not have the necessary specificity to tease out the role of SphK2-derived S1P, since ABC294640 also inhibits dihydroceramide desaturase in the de novo pathway of ceramide synthesis [31]. Our inability to identify a role for S1P in PGCC could be related to the function of this sphingolipid in promoting mitosis [32]. If S1P were to promote completion of cell division, PGCC could be driven into mitotic catastrophe, which suggests the possibility that survival in the PGCC state may necessitate a low level of intracellular S1P.

The LC/MS analysis demonstrated that saturated ceramides represent about 52–53% of all sphingoid bases in parental PPC1 but increases to 63.56% in PGCC. This difference disappears when cells are treated with LCL521, suggesting that ASAH1 activity is responsible for the altered balance of saturated ceramides but the significance of this observation remains to be elucidated. The LC/MS analysis also revealed that the distribution of ceramides species at steady state differs from the flux analysis. The flux assay demonstrated that sphingosine is primarily metabolized to C_16_-ceramide in the salvage pathway in PPC1 cells and in both PPC1 and MEL624 cells, C_16_-sphingolipids were the predominant species detected by LC/MS. Thus, C_16_-sphingolipids are abundant in these cells, yet intracellular C_16_-ceramide levels are relatively low suggesting that high levels of this species may adversely affect PGCCs. Since C_16_-ceramide has been associated with apoptosis, this lipid may be required to rapidly metabolized to complex forms to prevent cell death [33,34,35,36,37]. The high level of complex C_16_-sphingolipids permits cells to rapidly respond to stimuli and via activation of sphingomyelinases generate C_16_-ceramide upon demand. Our results of LC/MS analysis in the presence of LCL521 support this idea. We found that long chain ceramides, such as C_16_-ceramide, preferentially accumulate in PGCC when ASAH1 is inhibited by LCL-521 (Figure 1C). These ceramides could be generated de novo by ceramide synthases (CerS) or could arise through hydrolysis of ceramide from complex sphingolipids. However, we did not see any differences in the metabolism of ^17^C-Sph into ^17^C-ceramides between parental cancer cells and PGCC, suggesting that CerS activity in the salvage pathway is similar between the populations. Therefore, the increase in long chain ceramides is likely due to hydrolysis of sphingomyelin by sphingomyelinases (SMase) at the plasma membrane. Radiation has been shown to activate aSMase with liberated ceramide playing critical roles in raft formation and signal transduction [38]. nSMase2, which can be transcriptionally activated by p53, also liberates C_16_-ceramide from plasma membrane sphingomyelin when under stress [39,40]. SMases do not display substrate specificity but with C_16_-SM being the predominant form of sphingomyelin in both PPC1 and MEL624 cells, the activity of SMases would preferentially generate C_16_-ceramide. We previously demonstrated that C_16_-ceramide increases significantly following radiation or chemotherapy stress and leads to apoptosis in a majority of cells [13]. However, C_16_-ceramide can also transcriptionally activate ASAH1, suggesting that the function of ASAH1 in PGCC is to metabolize any pro-apoptotic C_16_-ceramide to sphingosine in order to survive. Taken together, our data support a model in which stress activated SMases preferentially generate C_16_-ceramide in the plasma membrane due to the high abundance of C_16_-SM. C_16_-ceramide either induces apoptosis or transcriptionally activates ASAH1 allowing a fraction of the population, primarily cells that are polyploid, to survive. ASAH1 supports PGCC functions through metabolizing excess C_16_-ceramide to sphingosine, which is rapidly recycled into ceramides and complex sphingolipid. Our results further suggest that PGCCs cycle C_16_-sphingolipids more rapidly than their parental counterparts, which may be an underlying reason for their increased dependence on ASAH1.

Given the abundance and apparent importance of C_16_-sphingolipids in PGCC, we next modulated levels of this sphingolipid species through expression of p53, which is stabilized by C_16_-ceramide and transcriptionally activates CerS6 the enzyme that preferentially generates C_16_-ceramide. We also targeted CerS6 directly through expression of an shRNA or overexpressed the enzyme directly using an adenoviral vector. Our results show that expression of p53 alone increases C_16_-ceramide levels and that the highest C_16_-ceramide levels were achieved when both CerS6 and p53 were expressed (Figure 3B). Our data also show that CerS6-derived C_16_-ceramide is important for p53 protein stability (Figure 3D), which supports the initial finding by the Krupenko lab that p53 is activated by C_16_-ceramide [41]. Although expression of p53 suppresses polyploidy at baseline, it does not alter the rate by which stress enhances polyploidy, suggesting that p53 expression does not change stress-induced polyploidy (Figure 4B). When equal numbers of PGCC are examined for their ability to generate progeny, expression of p53 results in significantly fewer colonies (compare Figure 4C,D). While modulation of CerS6 alone has no effect on polyploidy or progeny formation, expression of CerS6 is important for maximizing the suppressive effect of p53 (Figure 4D and Figure 5D). In MEL624 cells, which do not express appreciable amounts of CerS6 at baseline, p53 nevertheless increased C_16_-ceramide and reduced the ability of PGCC to generate progeny (Figure 5A–C). Expression of p53 alone increased C_16_-ceramide even in MEL624 cells, which lacked CerS6 expression. Nevertheless, CerS6 expression maximizes the adverse effect of p53 on progeny in PGCC, indicating that C_16_-ceramide regardless of source may be able to support p53 function. Taken together our results suggest that PGCC progeny can be prevented when C_16_-ceramide levels are high, which can be accomplished either through inhibition of ASAH1 or expression of p53. Expression of functional p53 maybe sufficient to prevent progeny formation in polyploid non-cancer cells. However, loss of p53 in cancer cells, especially in combination with aberrant sphingolipid metabolism that results in reduced C_16_-ceramides, favors progeny formation from PGCC.

## Figures and Tables

**Figure 1 cancers-13-02212-f001:**
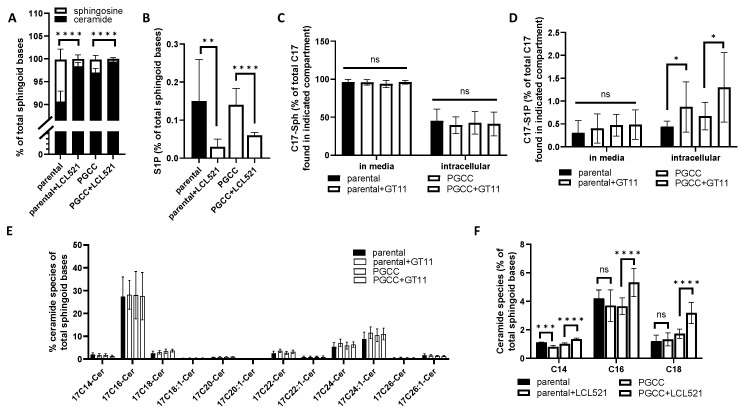
Steady-state and flux analysis of sphingolipids by LC/MS in parental and PGCC. (**A**) Steady-state LC/MS analysis to show differences sphingosine and ceramide content in PPC1 and PPC1-PGCC in the absence or presence of 5 μM LCL521. (**B**) Steady-state LC/MS analysis of S1P from data sets used in (**A**). (**C**–**E**). LC/MS analysis of ^17^C-Sph added for 30 min in the absence or presence of GT-11. (**C**) ^17^C-Sph detected in as a percentage of total in media or cells. (**D**) ^17^C-S1P detected in as a percentage of total in media or cells. (**E**) Distribution of ^17^C-ceramides detected in as a percentage of total intracellular ^17^C. (**F**) Steady-state LC/MS analysis of ceramide species in the absence or presence of 5 μM LCL521 for 5 h. Data are from three independently performed experiments each with triplicate samples. ANOVA was used to evaluate overall differences across groups with multiple conditions followed by Student’s *t*-test to further analyze specific differences. * *p* < 0.05, ** *p* < 0.01, *** *p* < 0.001, **** *p* < 0.0001. ns = not significant.

**Figure 2 cancers-13-02212-f002:**
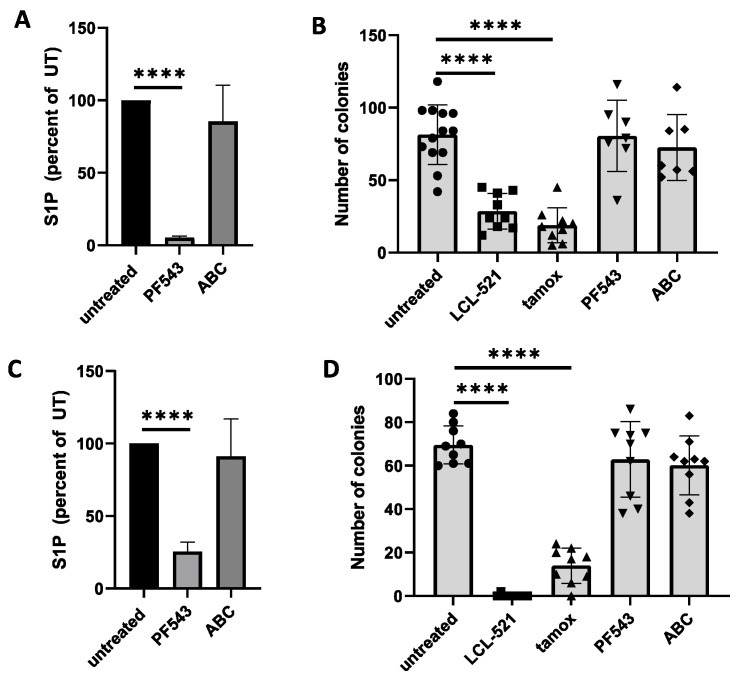
Effects of sphingosine kinase inhibition of S1P and colony formation from PGCC. S1P levels in the absence or presence of sphingosine kinase inhibitors in PPC1 (**A**) or MEL624 cells (**C**). Data are from two independently performed experiments with triplicate samples. Colony formation of PGCCs in the absence or presence of ASAH1 or sphingosine kinase inhibition in PPC1 (**B**) or MEL624 cells (**D**). Inhibitor concentrations were 5 μM for LCL521 or tamoxifen and 10 μM for PF543 or ABC294640. ANOVA was used to evaluate overall differences across groups with multiple conditions followed by Student’s *t*-test to further analyze specific differences. **** *p* < 0.0001. The symbols in panels (**B**,**D**) represent individual data points that were used to calculate the average and standard deviation.

**Figure 3 cancers-13-02212-f003:**
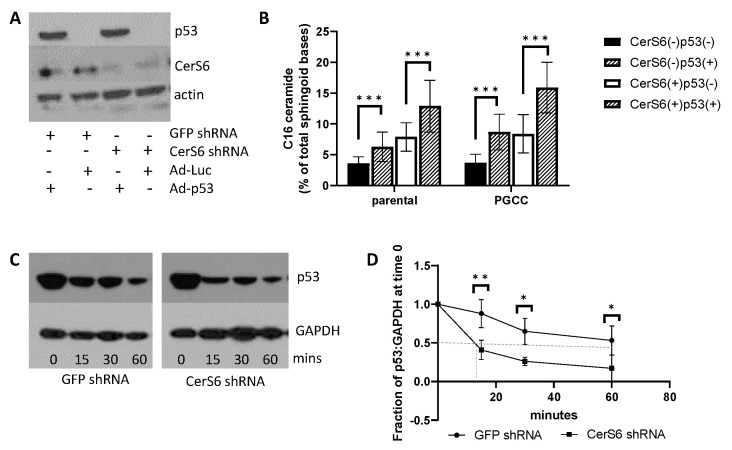
Expression and effects of CerS6 and p53 in PPC1 cells. (**A**) Western blot analysis of p53 and CerS6 expression following modulation by shRNA or viral transduction. (**B**) LC/MS analysis of C_16_-ceramide after altered expression of CerS6 and/or p53. (**C**) Representative Western blot analysis of p53 levels. (**D**) Densitometric analysis from five independent experiments with 50% expression indicated by the dotted line. Time on the X-axis indicates the duration of cycloheximide addition. ANOVA was used to evaluate overall differences across groups with multiple conditions followed by Student’s *t*-test to further analyze specific differences. * *p* < 0.05. ** *p* < 0.01, *** *p* < 0.001.

**Figure 4 cancers-13-02212-f004:**
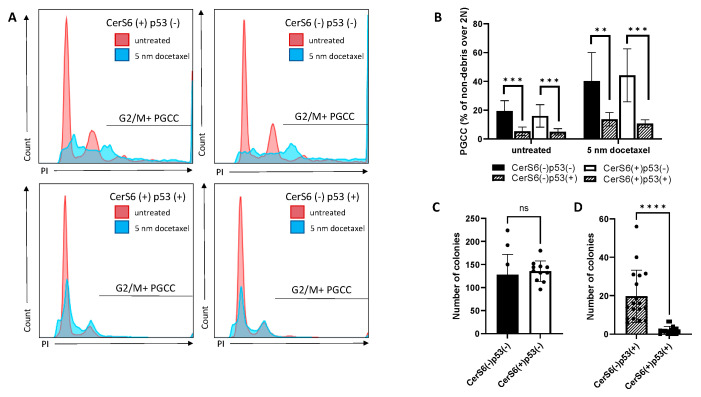
The effect of CerS6 and p53 modulation on polyploidy and PGCC progeny formation in PPC1 cells. (**A**) Flow cytometry traces of viable cells comparing parental cultures to PGCC-containing docetaxel treated cultures. (**B**) Quantification of (**A**). (**C**,**D**) Colony counts: 20,000 PGCC lacking (**C**) or expressing p53 (**D**) with modulated CerS6 expression. Data shown are from three independently performed experiments with multiple samples each. ANOVA was used to evaluate overall differences across groups with multiple conditions followed by Student’s *t*-test to further analyze specific differences. ** *p* < 0.01, *** *p* < 0.001, **** *p* < 0.0001.

**Figure 5 cancers-13-02212-f005:**
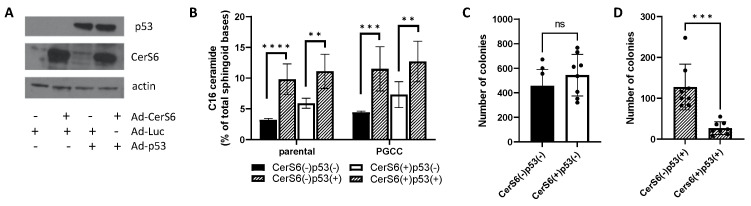
The effect of p53 and CerS6 expression in melanoma cells. (**A**) Western blot analysis of p53 and CerS6 expression following viral transduction. (**B**) LC/MS analysis of C_16_-ceramide after altered expression of CerS6 and/or p53. (**C**,**D**) Colony counts: 20,000 PGCC lacking (**C**) or expressing p53 (**D**) with modulated CerS6 expression. Data shown are from two independently performed experiments with multiple samples each. ANOVA was used to evaluate overall differences across groups with multiple conditions followed by Student’s *t*-test to further analyze specific differences. ** *p* < 0.01, *** *p* < 0.001, **** *p* < 0.0001.

## Data Availability

Data are available upon reasonable request.

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
