# Peer review of "Ceramide Synthase 6 Maximizes p53 Function to Prevent Progeny Formation from Polyploid Giant Cancer Cells"

_cancers, 2021, doi:10.3390/cancers13092212_

Round 1
Reviewer 1 Report
Criticism:
Figure 1B – are parental data +/- LCL compound really statistically significant? Looking at error bar it doesn’t seem to be the case;
Figure 1D, intracellular data- same concern as above;
White bars on Fig 1C, 1D, 1E and 1F looks absolutely same (i.e. empty inside) although the figure legend says they are different. That is also true for some other figures. This should be fixed;
I disagree with the statement on lanes 204-205 : “Thus inhibition of ASAH1 and subsequent loss of sphingosine would predict a decrease in C16-ceramide under these conditions, yet having observed the opposite suggested C16-ceramide is increased upstream of ASAH1.” In my opinion, not DECREASE but rather INCREASE of C16-ceramide could be expected upon ASAH1 inhibition UNLESS C16-ceramide is a poor substrate for ASAH1. Please clarify this;
Figure 2: Please indicate inhibitor concentrations in the Fig. 2 legend.
The data on lack of SphK2 involvement don’t look solid. The ABC294640 inhibitor didn’t show any effect in any experiments presented. Are you sure this inhibitor worked at all (perhaps it’s a bad lot)? Or perhaps SphK2 doesn’t express in cells you use? In addition, have you tested different concentrations of the ABC inhibitor- perhaps at the concentration used (10 micromoles?) was too low? Ideally a dose-response curve should be presented in such a case.
Please either: show that SphK2 doesn’t express in your cells, or if it does - find a working concentration of ABC inhibitor or if it doesn’t work - repeat experiments shown on Fig. 2 A,B with shRNA (or siRNA) against SphK2.
Lanes 236-237: “As positive controls, we included 5 µ M LCL-521 or 5 µM tamoxifen, which inhibit ASAH1 inhibition upstream of sphingosine kinases”. What does “inhibit ASAH1 inhibition” mean?
Figure 3B – are data, especially parental data +/- p53 statistically significant, especially with p<0.001? Looking at error bar it doesn’t seem to be the case. Please present raw numbers used for statistical analysis and the statistical calculations themselves.
Figure 3C- what do the 0-60 mins numbers mean-the length of treatment with cycloheximide? If yes, please state so in the figure legend and on panels C and D.
Authors are advised to repeat Figure 3C CerS6 shRNA Western normalizing protein level in each well by GAPDH loading -in the case p53 decrease will be much more convincing.
Figure 5A: CERS6 part looks originated from an experiment different from the one displaying p53 and actin levels. However, they all must be from the same experiment.
Discussion, lane 337: “but the significance of this observation remains to be eluded”. Eluded or elucidated?
Lanes 348-349: “We found that long chain ceramides, such as C16-ceramide, preferentially accumulate in PGCC when ASAH1 is inhibited by LCL521 (Fig 1C).” Figure 1C shows NO DIFFERENCE! If it’s Figure 1F, middle group of four bars, I am not sure about statistical significance of the data.
Overall, discussion is too long.
Reviewer 2 Report
The manuscript entitled, “Ceramide synthase 6 maximizes p53 function to prevent progeny formation from polyploid giant cancer cells,” from the Voelkel-Johnson group investigates the roles of sphingolipid alterations in the formation of PGCC and generation of progeny from them. The major observation of the study is that C16-ceramide, modulated via CerS6, regulates colony formation. This may be regulated by acid ceramidase, but not sphingosine kinases. These observations are of importance, but conclusions could be strengthened by further validation and assessment of specificity.
1. What is the specificity of the CerS6 response? Does CerS5 modulation exhibit a similar response? Do MEL624 cells express CerS5? As C18-ceramide is also increased (Fig. 1F), does CerS1 change progeny formation?
2. Showing the other ceramide species data in Figure 3B will add value by demonstrating if ceramides are undergoing remodeling vs increasing total ceramides by an increase in C16s. Fig. 3C would be improved by validating the shRNA knockdown of CerS6 and C16-ceramides levels.
3. The SMase pathway was suggested multiple times in the manuscript (e.g. Lines 222-224, 368). Additional studies ruling in/out SMase activation would help solidify the pathway(s) involved and how this may integrate with the proposed sphingosine recycling pathway.
Other comments:
1. For lipid normalization/assessment, further clarification is desired:
- Phosphate measurements – How was this done? Specifically, is this a lipid phosphorous measurement?
- With the exception of LCL treatment which lowered protein amounts, how do the experimental results compare between comparing to sphingoid bases versus protein?
- What sphingoid bases were used to derive % calculation? Were these hydrolyzed from the total sphingolipid pool?
Statistics – What post-hoc tests were used or was a t-test used in place of multiple comparisons?
While the authors point out specificity concerns of the ABC compound, it should be noted that isoform specificity is also suspect as concentrations often used in cell culture do not show isoform specificity by a target engagement assay (see PMID: 32835586). Likewise, the results of tamoxifen being attributed to being an ASAH1 inhibitor is weak. Conclusions would be better served with molecular strategies.
Line 172 – take out the word “the” between “….phosphorylated the downstream….”
Lines 370-371 – regarding excess C16-ceramide being converted to sphingosine being recycled back to ceramides and other SL. If the flux data points to generation of C16-ceramides, how does this suggest that it is supporting PGCC?
Reviewer 3 Report
Most of the anti-cancer therapies target mitosis and induce cell death, especially in malignant proliferative cells. Due to this stress, there are Polyploid giant cancer cells (PGCC) formed. These cells remain dormant for the most part but sometimes generate progeny via a primitive, cleavage-like cell division leading to a relapse of cancer spread. Studies show that this cell division is dependent on the sphingolipid enzyme acid ceramidase (ASAH1). The objective of this study described in the manuscript was to understand the molecular differences in sphingolipid metabolism between non-polyploid and polyploid cancer cells. This study will uncover the ASAH1-dependence in the PGCC population. The authors using steady-state and flux analysis of sphingolipids showed that sphingosine produced by ASAH1 is not converted to sphingosine-1-phosphate a pro-survival lipid signal. They showed that ASAH1 activity is required for preventing the accumulation of (C16-ceramide) long-chain ceramides.
Co-expression of p53 and CerS6 inhibited the ability of PGCC to form offspring, suggesting that both these genes are essential and form a positive feedback loop. CerS6 enhanced the effect of p53 by increasing half-life of the protein. This data suggests that identifying specific targets of sphingolipid metabolism that will inhibit the division and proliferation of PGCC population will be important to design better therapy. The authors have used the appropriate experimental methods and statistical in this study.
Author Response
The authors thank the reviewer for the positive comments.
Round 2
Reviewer 1 Report
Authors answered all of this reviewer's questions satisfactory. The only two things remains to be fixed are issue 3 - color or filling of bars in Figure 1, and Issue 13 - authors agreed it's Figure 1F should be mentioned in the Discussion rather than Figure 1C BUT DIDN'T CHANGE IT IN THE TEXT.
The paper has clinical significance and could be published after two minor fixes mentioned above.
Author Response
Thank you being so thorough in your review. We have modified the mention of Figure 1F on line 128 in the methods and on line 389 in the discussion.
As far as the issue with color not displaying correctly, we will have the editorial office verify because on our end the color displays correctly in both the Word and pdf versions of the manuscript.